



# Spatial- and temporal-patterns of global soil heterotrophic respiration in terrestrial ecosystems

Xiaolu Tang[1, 2], Shaohui Fan[3], Manyi Du[4], Wenjie Zhang[5, 6], Sicong Gao[6], Shibing Liu[1], Guo Chen[1], Zhen Yu[7], Yitong Yao[8], Wunian Yang[1]

[1]College of Earth Science, Chengdu University of Technology, Chengdu 610059, Sichuan, P.R. China

[2]State Environmental Protection Key Laboratory of Synergetic Control and Joint Remediation for Soil & Water Pollution, Chengdu University of Technology, Chengdu 610059, P. R. China

[3]Key laboratory of Bamboo and Rattan, International Centre for Bamboo and Rattan, Beijing 100102, P.R. China

[4]Experimental Center of Forestry in North China, Chinese Academy of Forestry, Beijing 102300, China

[5]State Key Laboratory of Resources and Environmental Information System, Institute of Geographic Sciences and Natural Resources Research, Beijing 100101, China

[6]School of Life Science, University of Technology Sydney, NSW 2007, Australia

[7]Department of Ecology, Evolution, and Organismal Biology, Iowa State University, Ames, IA 50011, USA

[8]Sino-French Institute for Earth System Science, College of Urban and Environmental Sciences, Peking University, Beijing 100871, P.R. China

*Correspondence to*: Shaohui Fan (fansh@icbr.ac.cn) ; Wunian Yang (ywn@cdut.edu.cn)

**Abstract.** Soil heterotrophic respiration (RH) is one of the largest and most uncertain components of the terrestrial carbon cycle, directly reflecting carbon loss from soil to the atmosphere. However, high variations and uncertainties of RH existing in global carbon cycling models require an urgent development of data-derived RH dataset. To fill this knowledge gap, this study applied Random Forest (RF) algorithm – a machine learning approach, to (1) develop a globally gridded RH dataset and (2) investigate its spatial- and temporal-patterns from 1980 to 2016 at the global scale by linking field observations from the Global Soil Respiration Database and global environmental drivers – temperature, precipitation, soil water content, etc. Finally, a globally gridded RH dataset was developed covering from 1980 to 2016 with a spatial resolution of half degree and a temporal resolution of one year. Globally, the average annual RH was $57.2 \pm 0.6$ Pg C a$^{-1}$ from 1980 to 2016, with a significantly increasing trend of $0.036 \pm 0.007$ Pg C a$^{-2}$. However, the temporal trend of the carbon loss from RH varied with climate zones that RH showed significant increasing trends in boreal and temperate areas, in contrast, such trend was absent in tropical regions. Temperature driven RH dominated 39% of global land and was mainly distributed at a high latitude. While the areas dominated by precipitation and soil water content were mainly semi-arid and tropical areas, accounting for 36% and 25% of



the global land, respectively, suggesting variations in the dominance of environmental controls on the spatial patterns of RH. The developed globally gridded RH dataset will further aid in understanding of the mechanisms of global soil carbon dynamics, serving as a benchmark to constrain global vegetation models. The dataset is publicly available at https://doi.org/10.6084/m9.figshare.8882567 (Tang et al., 2019a).

## 1 Introduction

Global soils and surface litter store up to 2- or 3-fold the amount of carbon present in the atmosphere (Trumbore, 2009) and therefore, a small change in soil carbon content could have profound effects on atmospheric $CO_2$ and climate change (Köchy et al., 2015). Although global carbon flux from soil-to-atmosphere is increasing (Zhao et al., 2017), the degree to which future climate change will stimulate soil carbon losses via heterotrophic respiration (RH) remains highly uncertain (Bond-Lamberty
et al., 2018; Friedlingstein et al., 2014; Trumbore and Czimczik, 2008), particularly in areas with high temperature sensitivity, rapid changes in precipitation frequency and intensity.

Soil RH represents carbon loss from the decomposition of litter detritus and soil organic matter by microorganisms, accounting for one of the largest components of the terrestrial carbon cycle (Bond-Lamberty et al., 2018). However, RH's feedback to climate variability is poorly understood. RH could affect future climate change via the mineralization of long-
stored soil carbon, offsetting net primary production (NPP) and even converting terrestrial ecosystems from carbon sink to carbon source (Tremblay et al., 2018). Conversion of the sink/source role depends on how strongly large-scale process affected by environmental drivers, e.g. temperature, precipitation and soil organic carbon content (Hursh et al., 2017; Sierra et al., 2015), or extreme conditions, such as fire, human disturbance and drought (Kurz et al., 2013; Metsaranta et al., 2011). Although it is widely recognized that warming enhances $CO_2$ release from soils, the magnitude of release is uncertain due to variations
in the temperature sensitivity of soil organic matter decomposition (Suseela et al., 2012). Therefore, reducing RH uncertainty and clarifying the response of RH to environmental factors are essential for future projections of the impact of climate change on the terrestrial carbon balance.

Due to the diurnal, seasonal and annual variability in RH, in addition to the difficulty of large-scale measurements, regional and global RH estimations mainly depend on modelling approaches using regional or global variables, such as temperature,
precipitation and carbon supply (Bond-Lamberty and Thomson, 2010b; Hashimoto et al., 2015; Hursh et al., 2017). However, to the best of our knowledge, no study has examined global RH variability using numerous empirical measurements that bridge the knowledge gap between local, regional and global scales spatially and temporally. In addition to temperature and precipitation, soil variables such as water, carbon and nitrogen contents are also important factors in the regulation of RH and should be considered for accurate RH estimations (Hursh et al., 2017; Zhao et al., 2017), although these variables vary with
biome and climate.



Observational studies have examined the responses of soil respiration to different climatic variables at different locations across the globe (Bond-Lamberty and Thomson, 2010a; Zhou et al., 2016). Hashimoto et al. (2015) and Bond-Lamberty and Thomson (2010b) predicted global soil respiration rates using climate-derived models driving by temperature and precipitation, however, the these models commonly explain less than 50% variations of soil respiration, requiring new techniques, potential

new numerical/algorithmic methods to better quantification and understanding of large-scale soil carbon flux (Bond-Lamberty, 2018). To improve the modelling accuracy, more recent studies have used linear regression or machine learning approaches including more abiotic or biotic variables, such as soil carbon supply, soil properties and NPP (Hursh et al., 2017; Zhao et al., 2017) and observations collected from newly published measurements (Jian et al., 2018; Zhao et al., 2017). On the other hand, including more variables in linear or non-linear regression models may cause overfitting and autocorrelation issues (Long and

Scott, 2006). To overcome overfitting and autocorrelation, machine learning approaches, such as the Random Forest (RF, Breiman, 2001), have been applied to explore the hierarchical importance of environmental factors, such as temperature, soil water content (SWC), NPP and soil pH (Hursh et al., 2017). Machine learning techniques are highly effective as they are fully data adaptive, and do not require initial assumptions on functional relationships and can function with nonlinear dependencies (Bodesheim et al., 2018). Therefore, these approaches have been widely used in earth science, particularly in carbon and water

flux modelling (Jung et al., 2010; Jung et al., 2017; Yao et al., 2018b), and may provide more reliable estimates of soil respiration (Bond-Lamberty, 2018; Zhao et al., 2017). However, no study to date has assessed the global variability of RH using empirically field observations to bridge the knowledge gap between local, regional and global scales.

The newly-emerged Dynamic Global Vegetation Model from the TRENDY model ensembles and Earth System Models have been widely used to investigate major physiological and ecological processes and ecosystem structures, providing a novel

database and approach to examine and estimate RH at the global scale (Zhu et al., 2017). TRENDY and Earth System Model simulations incorporating RH component are commonly calibrated and validated by eddy covariance measurements, e.g. net ecosystem carbon exchange (Yang et al., 2013), however, modelled RHs from these models have not yet been calibrated and validated using field RH observations. Therefore, these modelled RHs may be fundamentally different from observed values and no global observations exist to evaluate model effectiveness.

Thus, we used RF to estimate global RH based on updated RH observations from the Global Soil Respiration Dataset (SRDB, Bond-Lamberty and Thomson, 2010a) with the objectives of: (1) developing a globally gridded RH product (named data-derived RH); (2) detecting the temporal and spatial and temporal patterns of RH; (3) identifying the dominant driving factors for spatial- and temporal-variabilities of RHs; and (4) comparing data-derived RH dataset with data generated by Dynamic Global Vegetation Models from the TRENDY ensembles. Consequently, the developed global RH database could

improve our understanding the underlying mechanisms of RH variability to future climate change at the global scale, and could serve as a benchmark to constrain global vegetation models.

**2 Materials and methods**





## 2.1 Soil heterotrophic respiration database development

The basis of the database developed here included observed global RH values from SRDB (Bond-Lamberty and Thomson,
2010a), which was obtained at https://github.com/bpbond/srdb. The database was further updated using observations collected
from Chinese peer-review publications at the China Knowledge Resource Integrated Database (www.cnki.net) until March
2018. This study included the RH data for: (1) annual RH as directly reported in publications with at least one year continuous
measurements; (2) the start- and end-year were extracted from SRDB, directly from publications or calculated by the "years
of data" in the SRDB; (3) observations measured by alkali absorption or soda lime approaches were not included because of
their potential underestimation of respiration flux with increasing pressure in the measurement chamber (Pumpanen et al.,
2004); (4) experiments with treatments, such as nitrogen manipulation, or fertilization, were excluded, and only RH
measurements from the control treatment were included (Jian et al., 2018); (5) SRDB observations labelled as "potential
problem" ("Q10"), "suspected problem" ("Q11"), "known problem" ("Q12"), "duplicate" ("Q13") and "inconsistency"
("Q14") were not included (Bond-Lamberty and Thomson, 2010a). Finally, the newly updated dataset included 504
observations in total. Although most of the observations were from China, America and Europe, this database cover all the
major terrestrial biomes across the world (Fig. 1).

## 2.2 Climate and soil data

To investigate the global spatial-temporal RH patterns, global spatial-temporal grids of RH driving factors were required.
A total of 9 variables were included (Supplementary Table S1): Monthly gridded data of temperature, precipitation, diurnal
temperature range from Climatic Research Unit TS v.4.01 over 1901-2016 (https://crudata.uea.ac.uk, Harris et al., 2014);
shortwave radiation (SWR, https://www.esrl.noaa.gov, Kalnay et al., 1996); gridded soil organic carbon content (Hengl et al.,
2017)and nitrogen content from https://webmap.ornl.gov/ogc/index.jsp (Global Soil Data, 2000); monthly gridded nitrogen
deposition dataset from the global Earth System Models of GISS-E2-R, CCSM-CAM3.5 and GFDL-AM3 from 1850 to 2000s
(https://www.isimip.org, Lamarque et al., 2013); monthly Palmer Drought Severity Index (PDSI,
https://www.esrl.noaa.gov/psd/, Dai et al., 2004) and (SWC, https://www.esrl.noaa.gov, van den Dool et al., 2003). Before
further data analysis, monthly data were aggregated to an annual scale. These variables could stand for different environmental
controls on RH. For example, temperature, precipitation and SWC are critical environmental controls on microbial activities
for soil organic carbon decomposition (Jian et al., 2018; Suseela et al., 2012; Tremblay et al., 2018). Soil organic matter, soil
carbon stock and soil nitrogen are important carbon and nitrogen substrates for microbes that are related to the decomposition
of soil organic matter (Tremblay et al., 2018). The drought index (PDSI) and diurnal temperature range to represent water and
temperature stress on RH (Berryman et al., 2015; Zhu and Cheng, 2011). The global environmental drivers for each given site
were extracted by site longitudes and latitudes corresponding to annual RH observations. If the environmental driver is not in
a spatial resolution of 0.5º, we first re-sampled this environmental driver to 0.5º resolution using a bilinear interpolation.

## 2.3 RH from TRENDY models





In the last several decades, TRENDY models were developed to simulate key processes (e.g. photosynthesis, respiration, evapotranspiration, phenology and carbon allocation) that drive the dynamics of global terrestrial ecosystems (Piao et al., 2015). TRENDY models follow a common protocol and use the same climate-forcing data from National Centres for Environmental Prediction at a spatial resolution of 0.5 °. For model products with different spatial resolutions, new errors will be produced when re-sampling to 0.5 °. Therefore, to compare the dynamics in data-derived RH dataset and TRENDY RH

dataset, we used model outputs from seven TRENDY models (Community Land Model-4.5 (CLM4, Lawrence et al., 2011); Integrated Science Assessment Model (ISAM, Cao, 2005); Lund-Potsdam-Jena (LPJ, Sitch et al., 2003); Lund-Potsdam-Jena General Ecosystem Simulator (LPJ-GUESS, Smith et al., 2001); VEgetation-Global-Atmosphere-Soil (VEGAS, Zeng et al., 2005); and Vegetation Integrative Simulator for Trace gases (VISIT, Kato et al., 2013). Additionally, RH data generated empirically by Hashimoto et al. (2015) was compared, which was downloaded from the publicly available dataset

(http://cse.ffpri.affrc.go.jp/shojih/data/index.html) and estimated using empirical total soil respiration relationships using a climate-driven model (termed as Hashimoto RH) based on the observation from SRDB. More details can be found in Hashimoto et al. (2015).

### 2.4 RF-based RH Modelling

RF is a machine learning approach that uses a large number of ensemble regression trees, but a random selection of predictive

variables (Breiman, 2001). Two free parameter settings are required, which are the number of trees and candidate variables for each split. However, RF model is not usually sensitive to the number of trees or variables. RF regression can deal with a large number of features, assisting feature selection based on importance value of each variable and the avoidance of overfitting (Bodesheim et al., 2018; Jian et al., 2018). In the present study, RF model was trained using 9 variables(supplementary Table 1) in the *"caret"* package in R (R Core Team, 2018), which was then implemented to predict RH for each grid at a spatial

resolution of 0.5 °. To characterize the performance of RF, a 10-fold cross-validation was applied, which means that the dataset was stratified into 10 parts and each part contained roughly equal number of samples. The target values for each of these 10 parts were predicted based on training model using the remaining nine parts. Two model evaluation statistics were used, modelling efficiency and root mean square error  (RMSE, Tang et al., 2019b; Yao et al., 2018b).

### 2.5 Trend analysis

Trend analysis of RH were estimated by Theil-Sen linear regression and tested with the Mann-Kendall non-parametric test. The Theil-Sen estimator is a non-parametric slope estimator based on median values, and this approach was widely used for spatial analysis, such as time-series carbon flux (Dai et al., 2016) and vegetation greening and browning (Pan et al., 2018) . The Mann-Kendall non-parametric test was employed to investigate the significant changes in RH trend. The significance level was at 0.05.

**2.6 Relationships between RH and temperature, precipitation and SWC**




Although previous studies have used precipitation as a proxy for SWC (Bond-Lamberty and Thomson, 2010b; Chen et al., 2010), this may result in variability in soil respiration estimates (Jassal et al., 2007; Zhang et al., 2006), because the relationship between SWC and soil respiration is much more complex than that between soil respiration and temperature or precipitation (Jian et al., 2018). Therefore, the mean annual temperature (MAT), precipitation (MAP) and SWC were all considered as potentially important proxies driving RH (Bond-Lamberty et al., 2016; Reichstein and Beer, 2008). Annual mean RH was regressed against the three proxies. The relationships between data-derived RH and temperature/precipitation/SWC were assessed locally for each grid cell, by calculating the correlations using partial correlation analysis. When analysing the partial correlations between RH and the proxy, the other two proxies were controlled. The correlation strengths of temperature, precipitation and SWC were used to derive RGB combinations and indicate the drivers of RH.

## 2.7 The comparison map profile method

To detect the spatial similarity and difference patterns between data-derived RH and TRENDY or reported RH values from 1981 to 2010, we utilized the comparison map profile (CMP) method (Gaucherel et al., 2008). This method was based on the absolute distances (D) and cross-correlation coefficients (CC) across multiple scales, with D and CC reflecting the similarity and the spatial structures of two compared images with the same sizes, respectively (Gaucherel et al., 2008). The D value between moving windows (from 3 ×3 to 41 ×41 pixels in present study) of two compared images were calculated by Eq (1):

$$D = abs(\bar{x} - \bar{y}) \tag{1}$$

Where, $\bar{x}$ and $\bar{y}$ represent mean values calculated over two moving windows. Finally, the mean D value was calculated as an average of different moving windows.

The CC was calculated according to Eq (2):

$$CC = \frac{1}{N^2} \sum_{i=1}^{N} \sum_{j=1}^{N} \frac{(x_{ij} - \bar{x}) \times (y_{ij} - \bar{y})}{\sigma_x \times \sigma_y} \tag{2}$$

$$\sigma_x^2 = \frac{1}{N^2 - 1} \sum_{i=1}^{N} \sum_{j=1}^{N} (x_{ij} - \bar{x}) \tag{3}$$

Where, $x_{ij}$ and $y_{ij}$ are pixel values at $i^{th}$ row and $j^{th}$ column of the moving windows of two compared images, respectively; $N$ represents the total number of pixels covered by each of moving windows; $\sigma_x$ and $\sigma_y$ stand for standard deviations of two moving windows. Low D values reflect goodness between the compared images, while low CC values suggest low similarity. Finally, the mean D and CC were calculated as the averages from different moving windows.

## 3 Results

### 3.1 Spatial patterns of RH



Based on the 10-fold cross-validation, model efficiency ($R^2$) and *RMSE*, were 50% and 143 g C g C m$^{-2}$ a$^{-1}$ (Fig. S1), respectively. This indicates that the RF algorithm effectively captured the spatial- and temporal-variability of RH, therefore

enabling deriving of a global gridded RH dataset.

Data-derived RH dataset showed a strong spatial pattern globally (Fig. 2a). The largest RH fluxes occurred in tropical areas (e.g. Amazon tropical forest) at > 700 g C m$^{-2}$ a$^{-1}$, followed by the subtropics, such as South China and America, and humid temperate areas, e.g. North America, Western and Central Europe, with an annual RH of 400-600 g C m$^{-2}$ a$^{-1}$. Relatively low annual RH less than 200 g C m$^{-2}$ a$^{-1}$ was generally observed in areas with cold and dry climates, such as boreal areas,

characterized by low temperatures and short growing seasons, dry or semi-arid regions (e.g. Northwest China), where water availability limits ecosystem development. However, the most variable changes in RH - using standard deviation as a proxy (Fig. 2b), were found in boreal regions with more than 70 g C m$^{-2}$ a$^{-1}$, while the majority areas of RH variability exhibited smaller than 30 g C m$^{-2}$ a$^{-1}$. Similarly, TRENDY and previously reported RH values showed similar patterns with the highest RH in warm and humid areas and lowest RH in cold and dry regions (Fig. S2). However, differences existed in the absolute

RH fluxes (Fig. S2). For example, CLM4 and VISIT models predicted RH to be more than 1400 g C m$^{-2}$ a$^{-1}$ within Amazon forest regions, while ISAM and LPJ-GUESS estimates were typically low at around 1000 g C m$^{-2}$ a$^{-1}$. However, data-derived RH dataset and Hashimoto RH showed highest RH fluxes in tropical regions of less than 800 g C m$^{-2}$ a$^{-1}$.

To examine the similarity in patterns of data-derived RH dataset and those established by TRENDY models and Hashimoto RH, CMP method was employed (Fig. 3). Larger D and lower CC values indicate less consistent magnitudes and a local

gradient distribution between the two compared images. Data-derived RH dataset and Hashimoto RH differed greatly in East Canada and the Middle East with D values above 200 g C m$^{-2}$ a$^{-1}$ and CC values lower than -0.5. Interestingly, the most noticeable differences between the data-derived RH values and TRENDY model mean RH values, occurred in East Asia and the Middle East, where D was higher than 500 g C m$^{-2}$ a$^{-1}$, while CC was around -0.1. When assessing each TRENDY model individually (Figs. S3 and S4), the differences between data-derived RH dataset and TRENDY RH were even larger. The most

remarkable difference was found for CLM4 and VISIT models in regions where D was above 800 g C m$^{-2}$ a$^{-1}$ with CC values of about -0.3 (East Asia and America).

Across the latitudinal gradient, zonal mean RH values increased from cold or dry areas (e.g. tundra, and desert or semi-arid areas) to warm or humid areas (e.g. temperate and tropical areas, Fig. S5). Data-derived RH dataset varied from 60±12 at about 75°N to 640±71 g C m$^{-2}$ a$^{-1}$ at the equator, reflecting less stress from environmental limitations. In the dry tropical areas (10°S

-25°S and 10°N -25°N) limited by water, zonal mean RH decreased slightly. With the increase of water availability, RH showed a second peak in the Northern and Southern Hemispheres around 20°N and 40°S, respectively. However, there was a high level of variability between data-derived RH and those predicted by TRENDY/Hashimoto RH in equatorial regions (Fig. S5), with predictions generally overestimating RH at the equator. Peak RH values in the equatorial region ranged from 660±65g C m$^{-2}$ a$^{-1}$ in previously published estimations, to above 1200±460 g C m$^{-2}$ a$^{-1}$ for CLM4 model, resulting in a considerably higher

peak RH value for the model mean (950±300 g C m$^{-2}$ a$^{-1}$).





### 3.2 Total RH

Over the last 37 years, global RH has increased from 55.8 Pg C a$^{-1}$ (1 Pg = 1×10$^{15}$ g) in 1992 to 58.3 Pg C a$^{-1}$ in 2010, with an average of 57.2±0.6 Pg C a$^{-1}$ and strong annual variability (Fig. 4). Compared to the data-derived RH dataset, TRENDY/Hashimoto underestimated global RH values (Fig. 6a), with the exception of the VISIT model. ISAM predicted the

lowest global RH of 34.8±0.4 Pg C a$^{-1}$, while the VISIT model produced the highest RH of 59.9±0.6 Pg C a$^{-1}$ (Fig. 5a). The model mean RH was 47.6±0.5 Pg C a$^{-1}$, underestimating RH by 9.6 Pg C a$^{-1}$ (16%) in comparison to data-derived RH dataset. Due to this large divergence, the strength of correlation between data-derived RH and TRENDY/Hashimoto RH varied greatly from 0.06 to 0.72 (Fig. 5b). Boreal, temperate and tropical regions were the three most important contributors for global RH according to the Köppen – Geiger climate classification system (Peel et al., 2007), contributing 76% of the total global RH.

The mean RH of boreal, temperate and tropical areas were 10.8±0.3, 12.9±0.1 and 19.5±0.2 Pg C a$^{-1}$, accounting for 19%, 22% and 35% of total RH, respectively (Fig. S7).

### 3.3 Trends in RH

Globally, although there was a great inter-annual variability in RH, total RH has significantly increased at the rate of 0.036±0.007 Pg C a$^{-2}$ from 1980 to 2016 ($p$ = 0.000, Fig. 4). Comparison of data-derived RH dataset and that generated by

TRENDY models was performed, focusing on the period of 1981 to 2010. Data-derived RH increased at 0.041±0.01 Pg C a$^{-2}$ (Fig. S6), exhibiting a slower increasing trends of 0.05±0.007 and 0.057±0.009 Pg C a$^{-2}$ of TRENDY/Hashimoto RH, respectively. Additionally, temporal trends varied greatly among TRENDY models (Fig. S6), with the largest increasing trend of 0.123±0.013 Pg C a$^{-2}$ established by LPJ-GUESS and the largest decreasing trend of -0.018±0.007 Pg C a$^{-2}$ established by ISAM.

Temporal trends varied according to climate zone. RH in boreal and temperate areas increasing by 0.020±0.004 and 0.007±0.002 Pg C a$^{-2}$ from 1980 to 2016 ($p$ = 0.000, Fig. S7), while RH in tropical areas did not show a significant temporal trend, although inter-annual variabilities were observed ($p$ = 0.362, Fig. S7). TRENDY/Hashimoto RHs showed significant increasing temporal trends in boreal, temperate and tropical regions, except those estimated by ISAM and ORCHIDEE models (Fig. S8-10). However, the magnitude of increase varied among different TRENDY models.

From 1980 to 2016, global RH was expected to be driven by multiple environmental factors, such as temperature and precipitation. During this period, MAT and MAP levels increased significantly by 0.34±0.032°C and 6.69±2.399 mm per decade, respectively ($p$ < 0.01, Fig's. S11a and b). Therefore, the correlations between RH and temperature/precipitation were evaluated. Globally, RH was significantly correlated with temperature ($R^2$ = 0.56, $p$ = 0.000) and precipitation ($R^2$ = 0.42, $p$ = 0.000, Fig. 6) anomalies. On average, global RH increased by 1.08±0.163 Pg C a$^{-1}$ per 1°C increase in MAT and 0.23±0.046

Pg C a$^{-1}$ per 10 mm increment in MAP.

### 3.4 Spatial pattern of RH trends



Spatially, data-derived RH trends presented heterogeneous geographical patterns (Fig. 2c). Positively increasing trends in RH were found for more than half of the global land areas (59%, calculated from cell areas; Fig. S12). Generally, the increasing rates of RH were lower than 3 g C m$^{-2}$ a$^{-2}$, in contrast, the highest RH increase was above 6 g C m$^{-2}$ a$^{-2}$ in boreal regions, such

as Russia, North Canada and the Tibetan Plateau. RH exhibited a decreasing trend in 41% of the global land area and most considerably in South Asia (Fig. 2c). Similar to data-derived RH trends, RH trends estimated by the TRENDY/Hashimoto RH trend also showed heterogeneous geographical patterns (Fig. S13). However, large discrepancies were found among TRENDY/Hashimoto RH (Fig. 2c and S13). Generally, the largest increase in RH trends occurred in boreal areas, except for outputs by LPJ-GUESS and LPJ models, which showed a decreasing trend for most boreal areas. There was a decreasing trend

across most of tropics (e.g. Southeast Asia), with the exception of VEGAS model results (Fig. S13).

**3.5 Dominant factors in RH annual variability**

Annual mean temperature and precipitation were the most important factors dominating RH in 39% and 36% of global land area, respectively (Fig. S14). While SWC dominated the remaining 25% of global land area. Spatially, the dominant drivers controlling RH varied greatly across the globe (Fig. 2d), with the area dominated by temperature mainly distributed in boreal

areas above 50ºN. This was also observed in the relatively high and positive partial correlation coefficient between temperature and RH (Fig. S14a). While precipitation dominated temperate areas between 25ºN and 50ºN (such as North China, the Middle East and America), where a wide distribution of desert or semi-arid regions occur, SWC dominated in tropic areas, such the Amazon, India and Africa. Similarly, water availability (SWC and precipitation) were also main driving factors for RH in Australia.

Spatial patterns in environmental controls on TRENDY/Hashimoto RH varied greatly compared to data-derived RH dataset or among TRENDY models (Figs. 2d and S14-16). Water availability (including precipitation and SWC) appeared to be more important than temperature. The percentage of the areas dominated by temperature (mainly distributed in boreal areas, except for in ISAM model outputs), were less than the areas dominated by precipitation and SWC (globally distributed) (Fig. S16). In terms of the modelled mean RH, precipitation dominated in most global land areas (43%), followed by SWC (36%) and

temperature (21%) (Fig. S14).

**4 Discussion**

**4.1 Annual RH**

**4.1.1 Comparison with Hashimoto RH**

Despite the increasing efforts to quantify the global carbon cycle, large uncertainties still remain in the spatial and temporal

patterns in RH. To the best of our knowledge, this is the first study to apply the RF approach to predict of temporal and spatial patterns of global RH using field observations. Globally, mean RH amounted to 57.2±0.6 Pg C a$^{-1}$ from 1980 to 2016, 6.4 Pg C a$^{-1}$ higher than Hashimoto RH (Hashimoto et al., 2015). This difference may be due to several reasons, Firstly, the two RH



products covered different land areas, with data-derived RH dataset in the present study covering a higher land area. If data-derived RH dataset was masked by Hashimoto RH over 1981-2010, total RH was $51.8 \pm 0.6$ Pg C a$^{-1}$, close to that of Hashimoto RH with $51.1 \pm 0.7$ Pg C a$^{-1}$ (Fig. S17), respectively. However, the temporal and spatial patterns varied greatly (Figs. 3 and 5).

Secondly, the two RH products used different variables and algorithms for RH predictions. RH was not only affected by temperature and precipitation, but also by carbon substrates, soil nutrient levels and other variables (Hursh et al., 2017). Besides temperature and precipitation, we also included SWC, soil nitrogen and carbon contents as indicators for environmental and nutrient constraints on RH. Conversely, Hashimoto RH was estimated from a climate-driven model including only temperature and precipitation as the driving variables (Hashimoto et al., 2015). This simple model can partly explain the reasons that Hashimoto RH could not capture the significant decrease in RH in 1982 and 1991 due to El Chichón and Pinatubo eruptions, respectively (Zhu et al., 2016), while data-derived RH dataset and TRENDY RH successfully captured such effects.

Thirdly, the linear model between total soil respiration and RH was developed based on forest ecosystems (Bond-Lamberty et al., 2004; Hashimoto et al., 2015), which could be another uncertainty when applying this linear model to other ecosystems, e.g. croplands and grasslands.

### 4.1.2 Comparison with TRENDY models

As data-derived RH dataset often serve as a benchmark for global vegetation models, the empirically derived global patterns of annual RH was compared with TRENDY model results. Although data-derived RH dataset lied within the model range ($34.8 \pm 0.4$ Pg C a$^{-1}$ for ISAM to $59.9 \pm 0.6$ Pg C a$^{-1}$ for VISIT, Fig. 5a), the model mean underestimated RH by 16% compared to the data-derived RH dataset. Due to the different temporal trends among TRENDY model outputs and the low spatial correlation of data-derived RH dataset and TRENDY RH (correlation efficiencies ranging from 0.06 to 0.72, Fig. 6b), TRENDY RH clearly have different sensitivities to climate variability. Additionally, the difference in RH magnitude and spatial pattern varied considerably, as shown by analysis of absolute distances and cross-correlations. This effect was mostly notable in tropical areas in VISIT and CLM4 model outputs (Figs. S3 and S4). This phenomenon may be associated to several reasons. Firstly, plant functional types differed among TRENDY models. For example, the VEGAS model included four plant functional types (Zeng et al., 2005), while the LPJ model defined ten plant functional types (Sitch et al., 2003).

Secondly, for each set of equations, constant vegetation parameters (e.g. photosynthetic capacity) were applied across time and space for most TRENDY models, which may induce an RH bias. Model parameters using short-term observations do not account for inter-annual variability of climatic and soil conditions, generating a simplistic representation of RH due to the inability to capture the response of RH to new environmental controls in short-term observations.

Thirdly, models that do not consider nitrogen constraint could overestimate the increasing trend of RH, because nitrogen limitation was globally observed (LeBauer and Treseder, 2008). This could explain why the CLM4 model with nitrogen control produced a much smaller increasing trend compared to other TRENDY models, with the exception of ISAM (Fig. S6).





Therefore, including soil nitrogen as a driving variable in modelling RH in this dataset had the advantage to detect the nitrogen

constrain on RH.

Fourthly, the lacking of the representation of human activities and agricultural management (e.g. fertilization and irrigation) may underestimate RH, because fertilization and irrigation were important practices to increase RH (Chen et al., 2018; Zhou et al., 2016). This could explain why five of seven TRENDY models could not explain the significant increasing change of RH in middle China (Fig. S13), which experienced intensive use of fertilization for food security in recent decades.

Finally, uncertainties and differences in model structures could also lead to inconclusive RH estimations. Although the same climatic data, e.g. temperature and precipitation, were used for TRENDY models to reduce the uncertainty causing by various meteorological forcing, systematic errors may be caused by applying a particular forcing and the errors might be propagated to model outputs (Anav et al., 2015). Therefore, TRENDY models should be improved by incorporating more processes such as nutrient constrains and assessment of the model response to environmental variability (Keenan et al., 2012; Wang et al.,

2014; Yao et al., 2018b).

**4.2 Linkage to global carbon balance**

If assuming the global ratio of RH/total soil respiration ranged from 0.56 (Hashimoto et al. (2015)) to 0.63 (Bond-Lamberty et al. (2018)), annual soil respiration varied from 90.8 to 102.1 Pg C a$^{-1}$, within the reported values of 83 to 108 Pg C a$^{-1}$ based on recent studies (Bond-Lamberty and Thomson, 2010b; Hursh et al., 2017). This indirectly highlights the reliability of use of

RF for global RH prediction. Moreover, these findings also have relevance to carbon balance estimations. According to a recent NPP estimate from observations and IPCC report data (IPCC, 2013; Li et al., 2017), global NPP ranged from 61.5 to 60 Pg C a$^{-1}$, respectively. The residual between RH and NPP (net ecosystem production) was 2.8-4.3 Pg C a$^{-1}$, which is similar to global estimates of net ecosystem production from the International Geosphere-Biosphere Programme, which ranged from 1.9 to 4.1 Pg C a$^{-1}$ from 1959 to 2016 (Le Quéré et al., 2013; Le Quéré et al., 2016; Le Quéré et al., 2014).

With a 1°C increase in global MAT, RH will increase by 1.08 Pg C a$^{-1}$ globally, and it is 0.23 Pg C a$^{-1}$ for 10 mm increment in global MAP. These findings indicate that carbon flux from the decomposition of soil organic matter and litter (RH) maybe positively feedback to future global climate change - typically characterized by increasing temperature and alterations in precipitation (IPCC, 2013). However, this increment may not compensate for the carbon sink capability of terrestrial ecosystems, as the NPP increment per year (0.23 Pg C a$^{-1}$) was higher than that of RH (Li et al., 2017), suggesting an increased

carbon sink role for terrestrial ecosystems (Le Quéré et al., 2013).

**4.3 Dominant factors in RH**

Dominant factors driving RH varied spatially. As temperature and energy were the most limited climatic factor in high latitude areas, temperature was a dominant factor for RH in high latitudinal regions above 50°N (Fig. 2d), with low temperatures leading to low RH (Fig. 2a). Similarly, due to the limited amount of precipitation, RH in semi-arid areas was





mainly controlled by precipitation, which is consistent with reported both field observations (Bai et al., 2008) and modelling
studies (Gerten et al., 2008). SWC control of RH in tropical areas could be explained by the mechanisms of RH. Excessively
high SWC can reduce the diffusion of oxygen, while excessively low SWCs could limit water and soluble substrate availability,
preventing microbial activity (Luo and Zhou, 2006; Xu et al., 2004). Suseela et al. (2012) proposed that RH fluxes declined
sharply when volumetric soil moisture reduced below ~15% or exceeded ~26%, which supports the findings of the present
study. However, it should be noted that dominant environmental controls on spatial carbon flux gradients might vary among
different years (Reichstein et al., 2007), such as with climatic extremes.

**4.4 Temporal variability of tropical, temperate and boreal areas**

  Temporally, RH in tropical areas did not exhibit a temporal pattern between 1981 and 2010, indicating that the climate
change did not affect RH fluxes in these areas, negatively feeding back to future climate change. However, RH in boreal and
temperature areas experienced significant increasing trends of $0.020 \pm 0.004$ and $0.007 \pm 0.002$ Pg C $a^{-1}$, respectively (Fig. S6),
suggesting a positive feedback may occur with future climate change. Tremblay et al. (2018) proposed that increased RH was
mainly related to increasing temperatures in boreal forest soils, which supports the findings of the present study. It should be
noted that both data-derived RH dataset and Hashimoto/TRENDY RH in boreal areas showed a temporally increasing trend
from 1981 to 2010, although the magnitude of increase differed (Fig. S8). Furthermore, despite the ISAM model showing a
decreasing trend for temperate and tropic regions, the ISAM model had an increasing trend in RH from 1981 to 2010 in boreal
areas (Fig. S8). These results indicate that boreal regions are becoming increasingly important in global carbon cycling and
that the increasing trend may continue due to the large amount of carbon stored in soil. Therefore, climate change may
fundamentally alter carbon cycling in boreal areas through changes in the decomposition rate of soil organic matter (Crowther
et al., 2016; Hashimoto et al., 2015; Schuur et al., 2015). Furthermore, the response of RH to climate variability varied with
climate zone, indicating different carbon loss rates from RH will occur in different regions to future climate change.

**4.5 Advantages, limitations and uncertainties**

  Based on the updated SRDB dataset, we used RF algorithm to predict the temporal and spatial patterns of RH at the global
scale and its response to environmental variables, and empirically derived global patterns of annual RH could serve as a
benchmark for global vegetation models and reduce RH uncertainties. This developed RH dataset provides several advantages
to the estimation of global RH. Firstly, we compiled up-to-date field observations from SRDB and Chinese peer-review
literatures up to March 2018, including 504 observations in total covering the majority global terrestrial ecosystems and climate
zones (Fig. 1). Secondly, total RH and its inter-annual variability were assessed for boreal, temperate and tropical zones –
three main global climate zones. Analysis from data-derived RH dataset further concludes that RH in different climate zones
responded differently to global climate change. Thirdly, we applied RF to predict and map RH at the global scale using climate
and soil predictors. Compared to linear regression analysis for predicting soil respiration (as no such global RH predictions
were previously available for comparison), which has reported model efficiencies of <50% (Bond-Lamberty and Thomson,





2010b; Hashimoto et al., 2015; Hursh et al., 2017), RF algorithms achieve a higher model efficiency of 50%, allowing a feature selection according the importance value of each variable and avoiding overfitting (Bodesheim et al., 2018; Jian et al., 2018), improving RH modelling and reducing uncertainties. Additionally, data-derived RH dataset was cross-validated globally by
10-fold cross-validation (see "materials and methods" section), which could improve its reliability and feasibility compared to TRENDY based RH that are not validated and calibrated by field observations, bridging the knowledge gap between local, regional and global scales temporally and spatially with a large number of empirical field measurements.

However, although empirically derived global patterns of annual RH dataset could be used as a benchmark for the verification of global carbon cycle modelling, bridging the knowledge-gaps between local, regional and global scales, few
uncertainties and limitations still remained. Firstly, RF algorithms constructs a model based on training dataset and is typically data limited in terms of quantity, quality and representativeness. Uneven data distribution has been a known issue in many ecological studies across the world, e.g. Bond-Lamberty and Thomson (2010b), Jung et al. (2011) and Yao et al. (2018a). Uneven coverage of observations is an important source of uncertainty to develop the data-derived RH dataset, which cause a bias of RF model toward the areas with more observations. However, our dataset covered a large climatic and edaphic gradient
covering the major land covers and climate zones. Therefore, in future studies, increasing field observations in unsampled areas should greatly improve our ability to evaluate spatial and temporal patterns of RH at the global scale and model global carbon cycle to future climate change.

Secondly, the misrepresentation of human activities, particularly regarding to land management and land use change, could result in uncertainties in RH (Bond-Lamberty et al., 2016; Tang et al., 2016). These human activities include both site-level *in*
*situ* information and the corresponding global grids. Otherwise, such information must not be included as corresponding site information or respective globally gridded datasets are missing or insufficient. Although soil organic carbon stock, soil nitrogen content, SWC and shortwave radiation were selected as inputs for the development of the RF model, which could partly capture land use change, the impacts of land use change on inter-annual variability of RH have not been fully qualified in the present study. Therefore, further efforts are required to characterize and quantify the effects of land use changes on global RH.

Thirdly, the developed RH dataset was derived at an annual timescale, which may cause additional uncertainty regarding to the inter-annual variability of RH. Therefore, the need for a larger number of global observations and to develop finer-scale temporal dynamics need further exploration, in combination with remote-sensing measurements and field observations, which may provide new insights into terrestrial ecosystem carbon dynamics at the global scale.

## 5 Data availability

The developed globally gridded RH database and field RH observation dataset are publicly free for scientific purpose, which can be downloaded at https://doi.org/10.6084/m9.figshare.8882567 (Tang et al., 2019a). R codes to produce the main results are also available upon the request to first author or corresponding author.



## 6 Conclusions

Data-derived global RH dataset may be used as a benchmark for global vegetation models, however, no such study has yet
been conducted to assess the global variability in RH using a large dataset of empirical measurements to bridge the knowledge
gap between local, regional and global scales. To fill this knowledge gap, we developed a globally gridded RH dataset, which
was $0.5^{\circ} \times 0.5^{\circ}$ from 1980 to 2016 with an annually temporal resolution, using RF algorithm by linking field observations and
global variables. Robust conclusions include: (1) Annual mean RH was $57.2 \pm 0.6$ Pg C a$^{-1}$ between 1980 and 2016, with an
increasing trend of $0.036 \pm 0.007$ Pg C a$^{-2}$, indicating an increase in carbon loss from soil to atmosphere with future climate
change; (2) Significant temporal trends were observed in the RH in boreal and temperate areas, although none were found in
tropical regions. This indicates that the temporal trend in RH varied with climate zones, highlighting their different sensitivities
to future climate change; (3) The magnitude and dominant factors in RH results generated by data-derived and TRENDY
models varied greatly, indicating that future efforts should focus on improving the representation of RH in ecosystem models
and the ecosystem response to environmental variability; (4) More field observations are required in areas with limited
observational datasets, with the integration of smaller-scale temporal dynamics (rather than annual timescales) potentially
providing new insight into terrestrial ecosystem carbon dynamics at the global scale; (5) The develop globally gridded RH
dataset could serve as a benchmark to constrain the global vegetation models, further contributing to improve our
understanding of the mechanisms of global soil carbon dynamics.

**Author contributions.** XT, SF and WY design the study; XT, WZ, SG, MD, YY and ZY contributed to data analysis, including
improving R code; SL and GC provided constructive comments. All authors contributed to review the manuscript.

**Competing interests.** The authors declare that they have no conflict of interest.

**Acknowledgements.** This study was supported by Fundamental Research Funds of International Centre for Bamboo and
Rattan (1632017003 and 1632018003); Fundamental Research Funds of Public Welfare of Central Institutes
(CAFYBB2018MA002); the National Natural Science Foundation of China (31800365 and 41671432); Innovation funding of
Remote Sensing Science and Technology of Chengdu University of Technology (KYTD201501); Starting Funding of Chengdu
University of Technology (10912-2018KYQD-06910); Foundation for University Key Teacher of Chengdu University of
Technology (10912-2019JX-06910) and Open Funding from Key Laboratory of Geoscience Spatial Information Technology
of Ministry of Land and Resources (Chengdu University of Technology). Great thanks to Liang Liu, Yuhang Zhang and Xinrui
Luo for their kind help of data collection from Chinese publications and the contributors of global soil respiration dataset and



TRENDY models, Dai Palmer Drought Severity Index data provided by the NOAA/OAR/ESRL PSD, Boulder, Colorado, USA, from their Web site at https://www.esrl.noaa.gov/psd/.






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



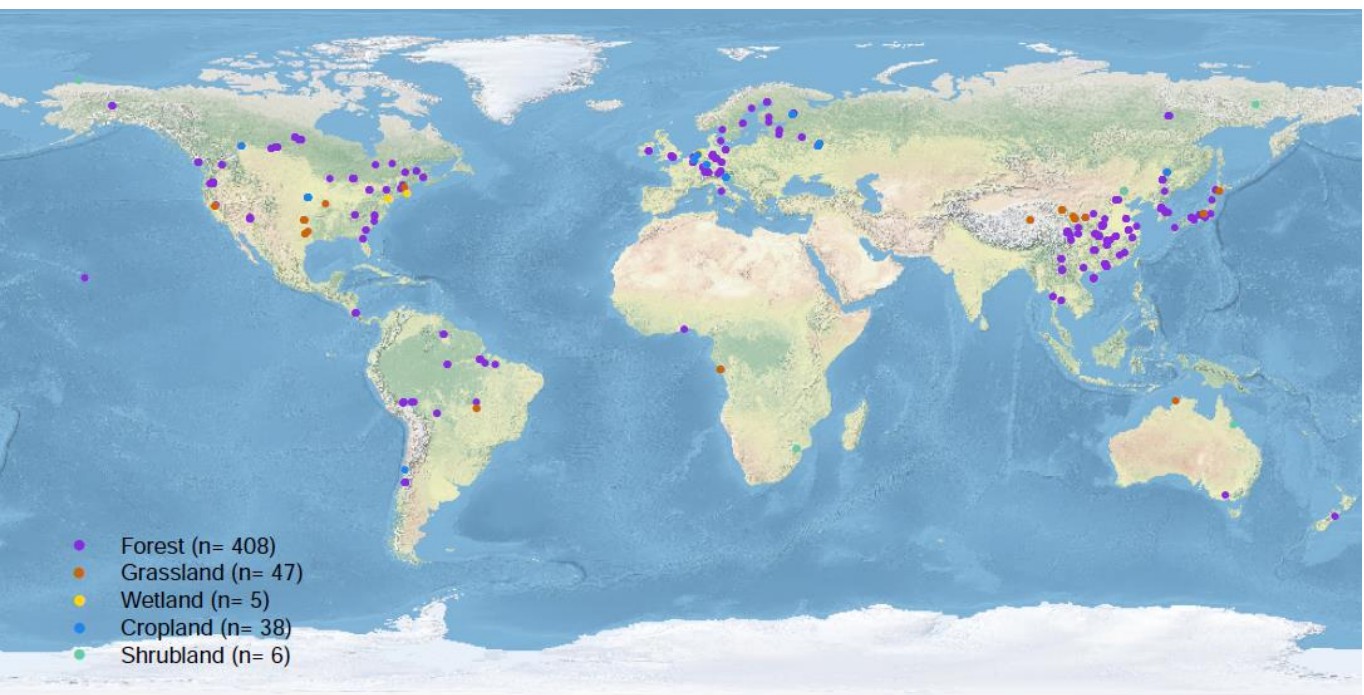

670          **Figure 1.** Distribution of the study sites for RH observations.



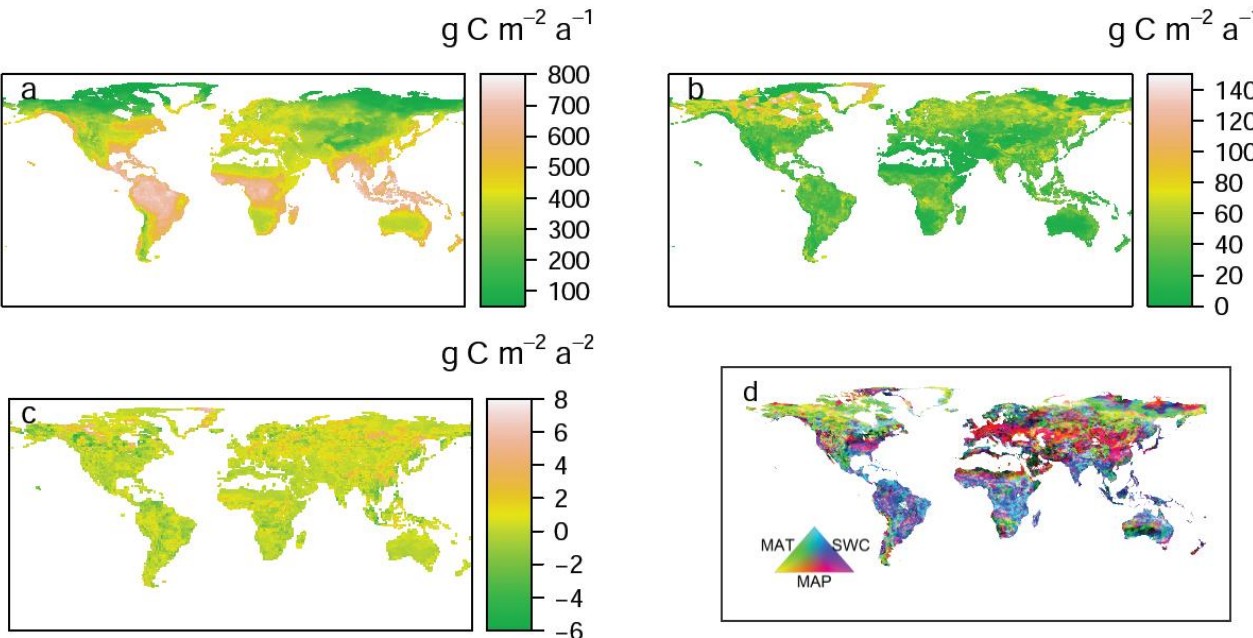

**Figure 2.** Spatial patterns of (a) mean data-derived RH (b) standard deviations, (c) temporal trends of annual heterotrophic respiration (RH) from 1980 to 2016, while (d) dominant environmental drivers for the inter-annual variability of global RH. MAT = mean annual temperature; MAP = mean annual precipitation; SWC = soil water content.

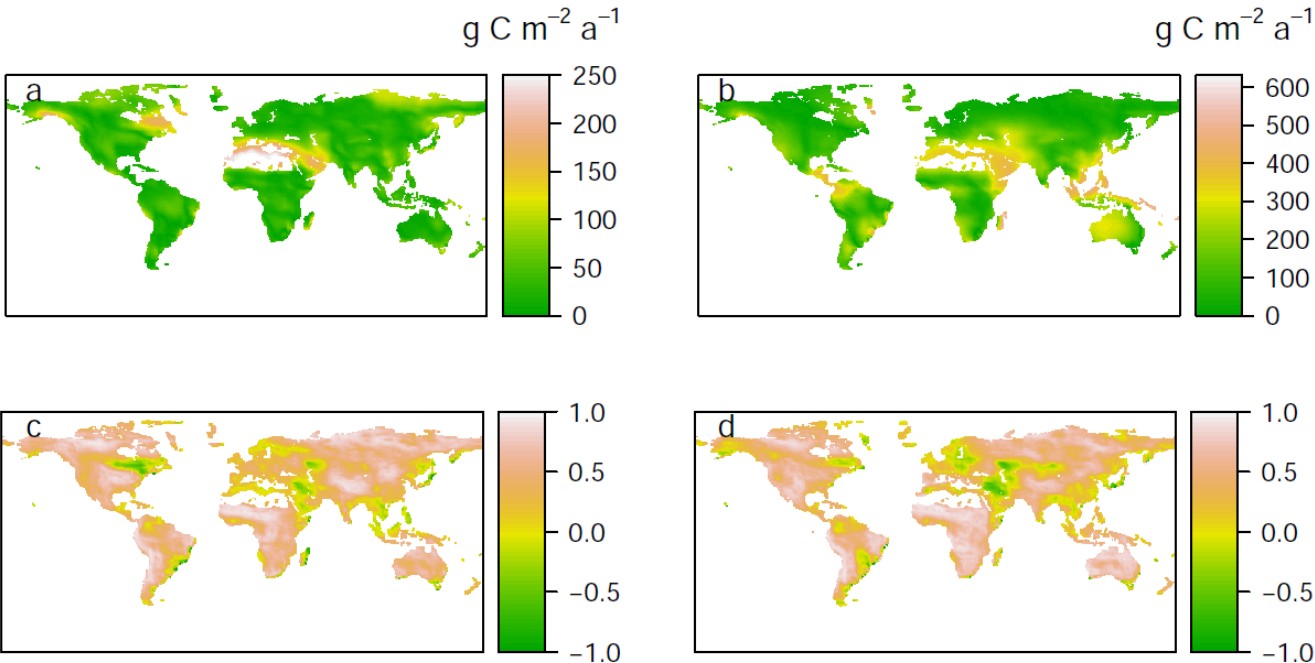

**Figure 3.** Comparing data-derived RH dataset with Hashimoto RH (a, c) and mean RH of TRENDY models (b, d) based on absolute distances (g C m$^{-2}$ a$^{-1}$, a, b) and cross-correlations (c, d). The absolute distances and cross-correlations were calculated using comparison map profile method (Gaucherel et al., 2008).



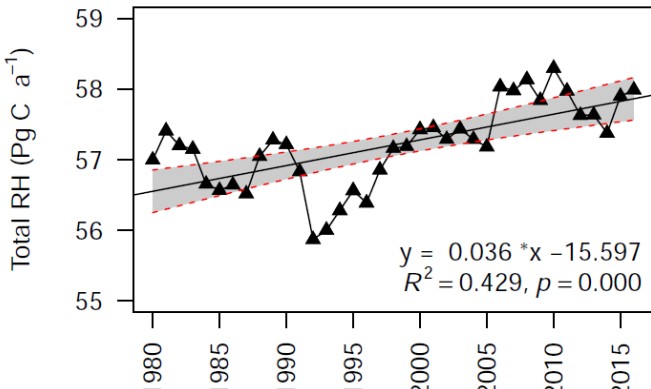

**Figure 4.** Inter-annual changes in global heterotrophic respiration (RH) from 1980 to 2016. The grey area indicates 95%
confidence intervals. For the linear regression model, $R^2 = 0.429$ and $p < 0.01$.





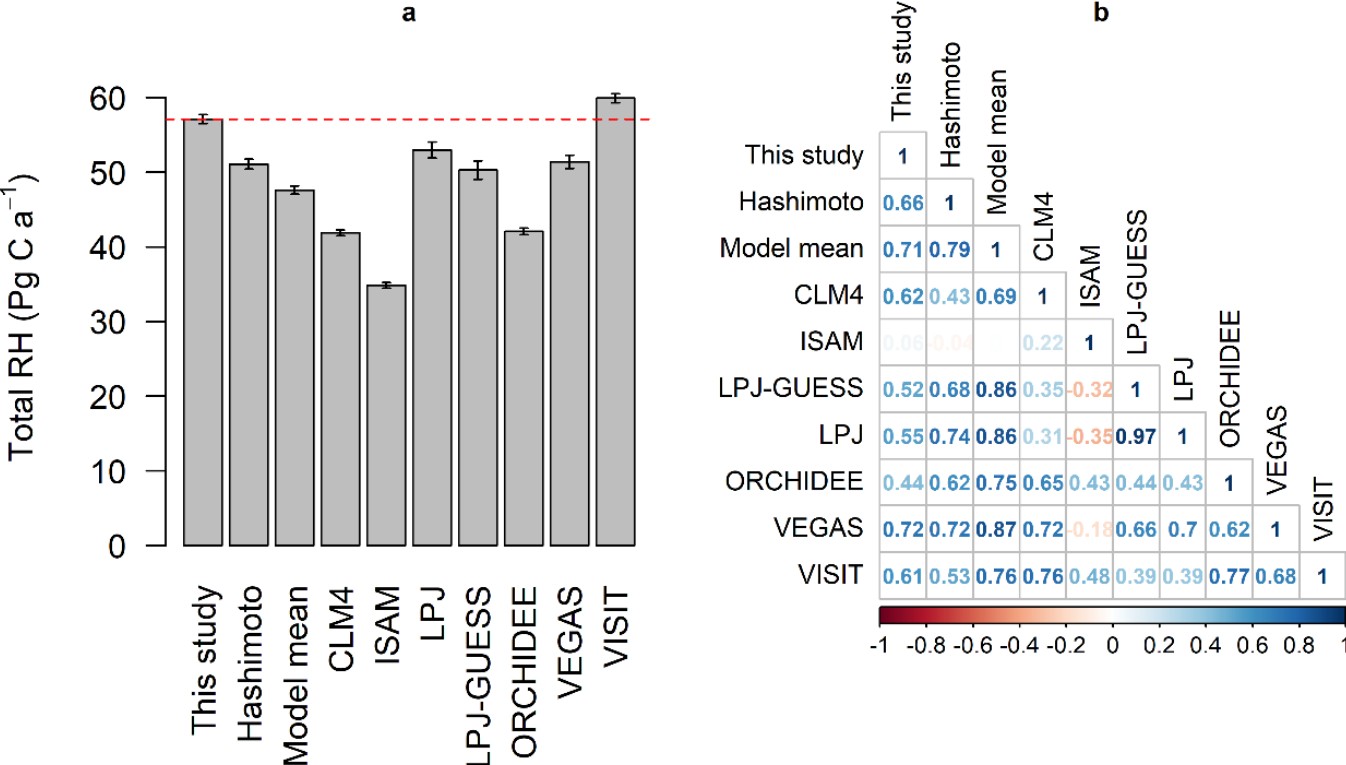

**Figure 5.** (a) Total global heterotrophic respiration (RH, mean ± standard deviation) fluxes and (b) the correlation coefficient analysed by Pearson correlation between data-derived RH and TRENDY /Hashimoto RH. The red dashed line (a) represents
690    the average of data-derived RH from 1981 to 2010.

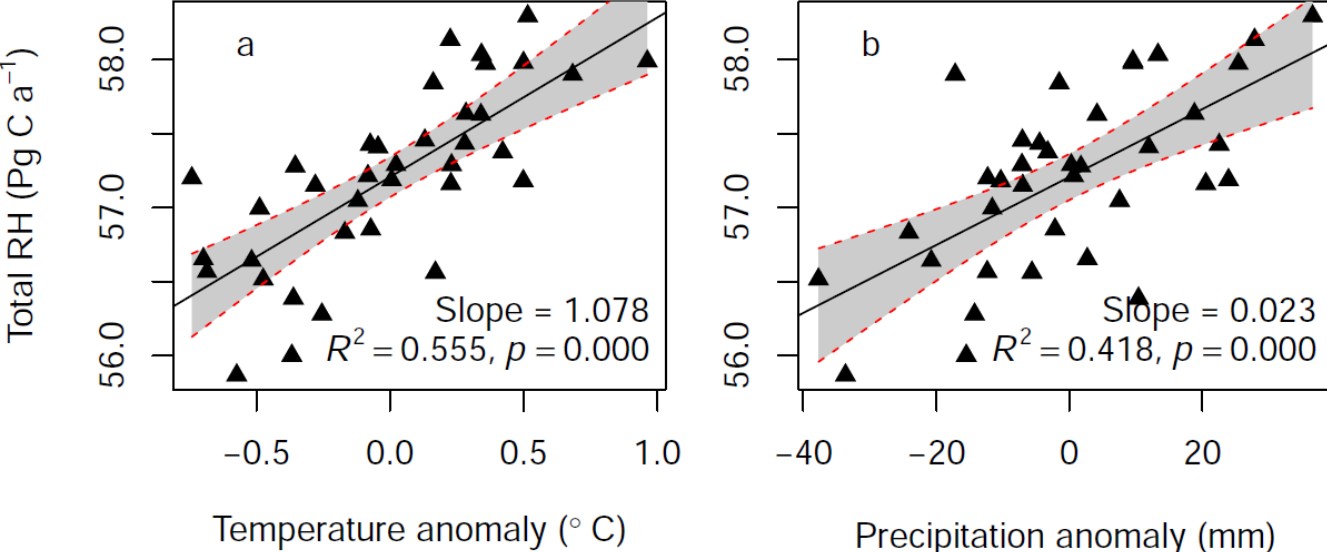

**Figure 6.** The relationships between heterotrophic respiration (RH) and mean annual temperature (a) or precipitation anomalies (b). The change was calculated as the difference of each given year to the average over 1980 to 2016. Grey areas
695  indicate 95% confidence intervals.