# Peer review of "Spatial- and temporal-patterns of global soil heterotrophic respiration in terrestrial ecosystems"

_Earth System Science Data, 2019_

## Referee Comment (RC1) · Ben Bond-Lamberty (Referee) · 5 Oct 2019

This interesting manuscript and dataset focus on global heterotrophic respiration (RH), using over 500 observations to produce a wall-to-wall global map over time. As the authors describe well, this is really important for understanding changes in the earth system, quantifying this poorly-constrained carbon flux, and benchmarking models. The ms is fairly well written and interesting; methods generally seem solid; references and discussion are well done.

There are some problems (see detailed list below). First, it's essential include the R code, and/or make it publicly available in a repository, for transparency and repro-

ducibility reasons; see below.

Second, although the discussion notes (395-) that one weakness of this dataset is the annual temporal resolution, it seems fair to note that a second is the half-degree spatial resolution. For many applications this is a significant limitation.

Finally, although the ms is very readable, there are many linguistic oddities and editing by a fluent English speaker would be useful.

Overall, this is an excellent study documenting a dataset that will be valuable for many carbon cycle and earth science researchers. It needs moderate revisions in a number of areas.

====================

Specific comments

1. Lines 43-44: remove this sentence perhaps? Awkward

2. L. 55-57 is basically repeated in 76-77; I'd remove it here

3. L. 74: perhaps "these approaches are beginning to be used" – the history of this is short

4. L. 80-: but note Shao et al. (2013), http://dx.doi.org/10.1088/1748-9326/8/3/034034

5. L. 89-91: this sentence probably should be moved to end of previous paragraph?

6. What software was used for all analyses? What versions (e.g. what version of caret)? Where is the code available/deposited? These are all critical for transparency and reproducibility

7. L. 183: the methods are a bit unclear–how was this R2 calculated? From the cross-validation?

8. L. 191-: changes over...? Space? Time? Clarify

9. L. 220: wow, that's a huge range

10. L. 276-: you should also address the 43.6 value calculated by Konings et al. (2019), http://dx.doi.org/10.5194/bg-16-2269-2019

11. L. 311-314: interesting idea!

12. L. 333-335: maybe note this is *really* uncertain though

13. L. 348: perhaps clarify to "indicating that in our model, climate change did not"

14. L. 356-357: yes! Perhaps put in abstract?

15. L. 382: Xu and Shang (2016), http://dx.doi.org/10.1016/j.jplph.2016.08.007, another really good reference here

16. L. 401-402: with respect, I think this (available on request) is not adequate; the R code should be deposited and open. See #6 above

17. Figure 1 could be improved: lat/lon references, marginal histograms

18. Figure 5: are the errors bars based on annual flux over many years? Clarify

---

## Referee Comment (RC2) · Ben Bond-Lamberty (Referee) · 6 Oct 2019

The dataset described by this manuscript, located at https://figshare.com/articles/Rh_720_360_1980_2016_nc/8882567, has some significant problems.

1. No units are given, either in the netcdf file or on the Figshare webpage.

2. Latitude appears to be inverted (i.e. mirrored around the equator).

3. If the units are in gC/m2/yr, as appears likely, note that the file includes no area variable (needed to scale to a global flux, for example).

[Figure]

4. Minor: RH is named "variable" in the file, and time is named "z", which is not particularly user-friendly.

---

## Referee Comment (RC3) · Anonymous Referee #2 · 6 Dec 2019

Heterotrophic respiration (RH) is a large component of the terrestrial carbon flux. There are large uncertainties in RH estimates, especially at the global scale. The study developed the global RH dataset using the RF method and RH observations. Overall, this manuscript is well written and interesting. The global dataset is valuable for global terrestrial carbon study. It is publishable after some modifications. RH is affected by numerous factors, including litter and soil carbon stocks, soil water content, soil temperature, and soil and litter properties. In this study, RH was estimated using the RF method driven by soil temperature, precipitation, and soil water content, soil organic carbon content. During the period from 1980 to 2016, soil carbon content might changes significantly due to the factor that the increase of productivity driven by $CO_2$

fertilization would induce the increase of litter input into soils. Without consideration of temporal variation of soil carbon content might induce uncertainties in the temporal trends of RH. Figure 1 shows that the number of sites used to train the RH model is limited and they are unevenly distributed. These sites are mainly located in America, Western Europe, and China. In Russia, Africa, Australia, Southwestern Asia, only data from few sites are available. It would be great if the uncertainty map of estimated RH can be provided. Double check the unit in Line 183.

---

## Author Comment (AC1) · 16 Dec 2019

**Response to reviewer 1**

Dear Dr. Bond-Lamberty,

Thank you very much for your encouragement, suggestion and comments. Based on your comments and suggestion, we revised our manuscript carefully and thoroughly. Please see, below, our point-to-point response.

**Suggestion and comments from referees** are marked in **Black**.

**Responses to referee's comments** are labelled in **blue**.

**Cited changes made in the manuscript** are marked in **red**.

Please do not hesitate to let us know if you have further questions and/or comments.

Sincerely,

Xiaolu Tang, Shaohui Fan and Wunian Yang, on behalf of all co-authors.

This interesting manuscript and dataset focus on global heterotrophic respiration (RH), using over 500 observations to produce a wall-to-wall global map over time. As the authors describe well, this is really important for understanding changes in the earth system, quantifying this poorly-constrained carbon flux, and benchmarking models. The ms is fairly well written and interesting; methods generally seem solid; references and discussion are well done. There are some problems (see detailed list below).

First, it's essential include the R code, and/or make it publicly available in a repository, for transparency and reproducibility reasons; see below.

Response: R codes are available on figureshare, and a new version of netcdf file include unit and area variables, was published as suggested. Please see the link: https://figshare.com/articles/Rh_720_360_1980_2016_nc/8882567

Second, although the discussion notes (395-) that one weakness of this dataset is the annual temporal resolution, it seems fair to note that a second is the half-degree spatial resolution. For many applications this is a significant limitation.

Response: thank you for your suggestion. We added this limitation in the discussion part as:

"Finally, we developed a global RH at a half-degree spatial resolution, which included a scale mismatch between the observations and global gridded variables. This could be a great challenge for spatial modelling and using global gridded variables with a finer resolution is encouraged to overcome this limitation (Xu and Shang, 2016). On the other hand, the study sites were globally distributed and there was a large climatic and edaphic gradient covering the major land covers and biomes, which should reflect a larger variability than the site-to-grid mismatch."

Finally, although the ms is very readable, there are many linguistic oddities and editing by a fluent English speaker would be useful. Overall, this is an excellent study documenting a dataset that will be valuable for many carbon cycle and earth science researchers. It needs moderate revisions in a number of areas.

Response: thank you. Our manuscript has already sent to a company for language improvement. Besides, we carefully revised the language again. See the language changes in revised version (with track change) and the language proof certification below:

[Figure]

**CERTIFICATE OF ENGLISH EDITING**

This is to certify that the manuscript entitled

**_Spatial- and temporal-patterns of global soil heterotrophic respiration in terrestrial ecosystems_**

commissioned to us has been carefully edited by a native English-speaking editor of MogoEdit, and the grammar, spelling, and punctuation have been verified and corrected where needed. Based on this review, we believe that the language in this paper meets academic journal requirements. Please contact us with any questions.

[Figure]

*Gang Zhang*

Dr. Gang Zhang
Founder & CEO of MogoEdit

Date of Issue
April 19, 2019

**Disclaimer:** Subsequent to our editing, a manuscript will be reviewed by the author(s) and then carefully rechecked by our editors during a second round of editing prior to submission. This manuscript however, received one round of editing only. The suggested edits in the document may therefore have been accepted or rejected by the authors at their sole discretion subsequent to our editing. Consequently, MogoEdit is not responsible for revisions made to the document after our last edit on **April 19, 2019**.

MogoEdit is a professional English editing company who provides English language editing, translation, and publication support services to individuals and corporate customers worldwide. As a company invested by the affiliate fund of Chinese Academy of Science, MogoEdit is one of the leading language editing service providers in China, whose clients come from more than 1000 universities and research institutes.
MogoEdit Website:   http://en.mogoedit.com/
500+ native English editors:   http://en.mogoedit.com/editors

[Figure]

Mogo Internet Technology Co., LTD.
No. 57, 3rd Keji Road, Xi'an 710075, PR China +86 02988317483     support@mogoedit.com

Specific comments

1. Lines 43-44: remove this sentence perhaps? Awkward

Response: this sentence is the definition of RH, which could be important for readers to understand the manuscript. We kept this sentence and cited more appropriate references.

"Soil RH represents carbon loss from the decomposition of litter detritus and soil organic matter by microorganisms (Subke et al., 2006), accounting for one of the largest components of the terrestrial carbon cycle (Bond-Lamberty et al., 2016)"

2. L. 55-57 is basically repeated in 76-77; I'd remove it here

Response: we removed this sentence.

3. L. 74: perhaps "these approaches are beginning to be used" – the history of this is short

Response: we corrected this sentence as:

"Therefore, these approaches are beginning used in earth science"

4. L. 80-: but note Shao et al. (2013), http://dx.doi.org/10.1088/1748-9326/8/3/034034

Response: we agree and proposed that:

"although RH improvements in Earth System Models are required (Shao et al., 2013)"

5. L. 89-91: this sentence probably should be moved to end of previous paragraph?

Response: done!

6. What software was used for all analyses? What versions (e.g. what version of caret)? Where is the code available/deposited? These are all critical for transparency and reproducibility

Response: all data analysis were conducted in R and R codes are available at figureshare: https://figshare.com/articles/Rh_720_360_1980_2016_nc/8882567.

Caret version 6.0-80 (accessed on May 27, 2018), was used.

7. L. 183: the methods are a bit unclear–how was this R2 calculated? From the crossvalidation?

Response: yes, R2 means model efficiency, which was built from the 10-fold cross-validation.

8. L. 191-: changes over: : :? Space? Time? Clarify

Response: sorry, we mean "change over the time from 1980 to 2016".

"However, the most variable changes in RH over the time from 1980 to 2016"

9. L. 220: wow, that's a huge range

Response: yes, there was a huge range.

10. L. 276-: you should also address the 43.6 value calculated by Konings et al. (2019), http://dx.doi.org/10.5194/bg-16-2269-2019

Response: done! We cited the reference and compared the result.

"Globally, mean RH amounted to 57.2±0.6 Pg C a-1 from 1980 to 2016, 13.6 Pg C a-1 than RH from satellite-driven estimates (Konings et al., 2019) , and 6.4 Pg C a-1 higher than Hashimoto RH (Hashimoto et al., 2015). The differences between data-driven RH in this study and Hashimoto may be due to several reasons"

11. L. 311-314: interesting idea!

Response: thanks!

12. L. 333-335: maybe note this is *really* uncertain though

Response: we remove this sentence.

13. L. 348: perhaps clarify to "indicating that in our model, climate change did not"

Response: done!

14. L. 356-357: yes! Perhaps put in abstract?

Response: we have already stated that RH in boreal area increased over 1980 – 2016. Therefore, to avoid the repetition, we did not add this sentence in abstract.

15. L. 382: Xu and Shang (2016), http://dx.doi.org/10.1016/j.jplph.2016.08.007 , another really good reference here

Response: thanks and we cited this references as well.

16. L. 401-402: with respect, I think this (available on request) is not adequate; the R code should be deposited and open. See #6 above

Response: we added R code to the figureshare:

https://figshare.com/articles/Rh_720_360_1980_2016_nc/8882567

17. Figure 1 could be improved: lat/lon references, marginal histograms

Response: we improved the figure and added lat/lon references and changed the marginal as follows:

[Figure]

Fig. 1 Distributions of the study sites for RH observations.

18. Figure 5: are the errors bars based on annual flux over many years? Clarify

Response: we meant "standard deviation of annual RH from 1981 to 2010" and added this information in figure caption.

References:

Bond-Lamberty, B., Epron, D., Harden, J., Harmon, M. E., Hoffman, F., Kumar, J., McGuire, A. D., and Vargas, R.: Estimating heterotrophic respiration at large scales: challenges, approaches, and next steps, Ecosphere, 7, e01380, http://dx.doi.org/10.1002/ecs2.1380, 2016.

Hashimoto, S., Carvalhais, N., Ito, A., Migliavacca, M., Nishina, K., and Reichstein, M.: Global spatiotemporal distribution of soil respiration modeled using a global database, Biogeosciences, 12, 4121–4132, http://dx.doi.org/10.5194/bgd-12-4331-2015, 2015.

Konings, A. G., Bloom, A. A., Liu, J., Parazoo, N. C., Schimel, D. S., and Bowman, K. W.: Global satellite-driven estimates of heterotrophic respiration, Biogeosciences, 16, 2269-2284, http://dx.doi.org/10.5194/bg-16-2269-2019, 2019.

Shao, P., Zeng, X., Moore, D. J. P., and Zeng, X.: Soil microbial respiration from observations and Earth System Models, Environmental Research Letters, 8, 034034, http://dx.doi.org/10.1088/1748-9326/8/3/034034, 2013.

Subke, J.-A., Inglima, I., and Francesca Cotrufo, M.: Trends and methodological impacts in soil $CO_2$ efflux partitioning: A metaanalytical review, Glob. Chang. Biol., 12, 921-943, http://dx.doi.org/10.1111/j.1365-2486.2006.01117.x, 2006.

Xu, M. and Shang, H.: Contribution of soil respiration to the global carbon equation, J. Plant Physiol., 203, 16-28, https://doi.org/10.1016/j.jplph.2016.08.007, 2016.

---

## Author Comment (AC2) · 16 Dec 2019

**Response to reviewer 1**

Dear Dr. Bond-Lamberty,

Thank you very much for your suggestion and comments to improve the netcdf dataset. Based on your comments and suggestion, we revised the netcdf dataset carefully. Please see, below, our point-to-point response and changes in the netcdf dataset at: https://figshare.com/articles/Rh_720_360_1980_2016_nc/8882567.

Please do not hesitate to let us know if you have additional questions and/or comments.

Sincerely,

Xiaolu Tang, Shaohui Fan and Wunian Yang, on behalf of all co-authors.

The dataset described by this manuscript, located at

https://figshare.com/articles/Rh_720_360_1980_2016_nc/8882567, has some significant problems.

1. No units are given, either in the netcdf file or on the Figshare webpage.

Response: sorry! We added the unit (g C m$^{-2}$ yr$^{-1}$) both in Figshare webpage and the netcdf. Please see https://figshare.com/articles/Rh_720_360_1980_2016_nc/8882567:

2. Latitude appears to be inverted (i.e. mirrored around the equator).

Response: we checked the netcdf file using "brick" and "spplot" function in raster package in R (version 3.5.0) and Latitude was not inverted.

3. If the units are in gC/m2/yr, as appears likely, note that the file includes no area variable (needed to scale to a global flux, for example).

Response: the unit (g C m$^{-2}$ yr$^{-1}$) both in Figshare webpage and the netcdf. A new file named "land area" in this study was upload to the figureshare as well.

4. Minor: RH is named "variable" in the file, and time is named "z", which is not particularly user-friendly.

Response: sorry! The variable in netcdf file was renamed "RH", while time was renamed "Year". Please see https://figshare.com/articles/Rh_720_360_1980_2016_nc/8882567.

---

## Author Comment (AC3) · 16 Dec 2019

**Response to Reviewer 2**

Dear reviewer,

Thank you very much for your great efforts, comments and suggestion! Based on the comments and suggestion, we revised our manuscript carefully and thoroughly. Please see, below, our point-to-point response.

**Suggestion and comments from referees** are marked in **Black**.

**Responses to referee's comments** are labelled in **blue**.

**Cited changes made in the manuscript** are marked in **red**.

Please do not hesitate to let us know if you have further questions and/or comments.

Sincerely,

Xiaolu Tang, Shaohui Fan and Wunian Yang, on behalf of all co-authors.

Heterotrophic respiration (RH) is a large component of the terrestrial carbon flux. There are large uncertainties in RH estimates, especially at the global scale. The study developed the global RH dataset using the RF method and RH observations. Overall, this manuscript is well written and interesting. The global dataset is valuable for global terrestrial carbon study. It is publishable after some modifications.

RH is affected by numerous factors, including litter and soil carbon stocks, soil water content, soil temperature, and soil and litter properties. In this study, RH was estimated using the RF method driven by soil temperature, precipitation, and soil water content, soil organic carbon content. During the period from 1980 to 2016, soil carbon content might changes significantly due to the factor that the increase of productivity driven by CO2 fertilization would induce the increase of litter input into soils. Without consideration of temporal variation of soil carbon content might induce uncertainties in the temporal trends of RH. Figure 1 shows that the number of sites used to train the RH model is limited and they are unevenly distributed. These sites are mainly located in America, Western Europe, and China. In Russia, Africa, Australia, Southwestern Asia, only data from few sites are available.

Response: we full agree with the reviewer (1): soil carbon content might change from 1980 to 2016 and (2) Uneven data distribution has been a known issue that may cause uncertainly to the areas without observations, and we added these limitations in the discussion part - Advantages, limitations and uncertainties, as:

"Besides, without consideration of the temporal changes of soil organic carbon content from 1980 to 2016 might bring uncertainties because the increase of productivity driven by $CO_2$ fertilization would increase litter input into soils. However, there is a lack of soil organic carbon content that considering its temporal changes based on observations, which has constrained the further analysis of the effects of the temporal changes of soil organic carbon content on RH."

"Uneven data distribution has been a known issue in many ecological studies across the world, e.g. Bond-Lamberty and Thomson (2010), Jung et al. (2011), Xu and Shang (2016) and Yao et al. (2018). The observations were mainly from China, Europe and North America, while there were a lack of observations in Russia, Africa, Australia and Southwestern Asia in our study."

It would be great if the uncertainty map of estimated RH can be provided.

Response: thank you for your good suggestion. We produced an uncertainty map defined as the ratio of standard error and mean value according to Greaves et al. (2016). We added this uncertainty map in the supplementary file and cited in manuscript:

"However, the most variable changes in RH over the time from 1980 to 2016 - using standard deviation and coefficient of variation (CV, the ratio of the standard deviation and the mean) as proxies (Fig. 2b an S3), were found in boreal regions with more than 70 g C $m^{-2}$ $a^{-1}$ or CV > 0.7, while the majority areas of RH variability exhibited smaller than 30 g C $m^{-2}$ $a^{-1}$ or CV < 0.3."

[Figure]

Fig. S3 The uncertainty map (the ratio of the standard deviation and the mean) of RH from 1980 to 2016 following Greaves et al. (2016).

Double check the unit in Line 183.

Response: thank you for your careful revision. We changed the unit to "g C m$^{-2}$ a$^{-1}$"

References:

Bond-Lamberty, B. and Thomson, A.: Temperature-associated increases in the global soil respiration record, Nature, 464, 579-582, http://dx.doi.org/10.1038/nature08930, 2010.

Greaves, H. E., Vierling, L. A., Eitel, J. U. H., Boelman, N. T., Magney, T. S., Prager, C. M., and Griffin, K. L.: High-resolution mapping of aboveground shrub biomass in Arctic tundra using airborne lidar and imagery, Remote Sens. Environ., 184, 361-373, https://doi.org/10.1016/j.rse.2016.07.026, 2016.

Jung, M., Reichstein, M., Margolis, H. A., Cescatti, A., Richardson, A. D., Arain, M. A., Arneth, A., Bernhofer, C., Bonal, D., Chen, J. Q., Gianelle, D., Gobron, N., Kiely, G., Kutsch, W., Lasslop, G., Law, B. E., Lindroth, A., Merbold, L., Montagnani, L., Moors, E. J., Papale, D., Sottocornola, M., Vaccari, F., and Williams, C.: Global patterns of land-atmosphere fluxes of carbon dioxide, latent heat, and sensible heat derived from eddy covariance, satellite, and meteorological observations, J. Geophys. Res. Biogeosci., 116, G00J07, http://dx.doi.org/10.1029/2010jg001566, 2011.

Xu, M. and Shang, H.: Contribution of soil respiration to the global carbon equation, J. Plant Physiol., 203, 16-28, https://doi.org/10.1016/j.jplph.2016.08.007, 2016.

Yao, Y., Piao, S., and Wang, T.: Future biomass carbon sequestration capacity of Chinese forests, Science Bulletin, 63, 1108-1117, http://dx.doi.org/10.1016/j.scib.2018.07.015, 2018.

---

## Author Response (AR2)

**Cover letter**

Dear editor and reviewer,

Thank you very much for your great efforts, comments and suggestion! According to the third reviewer's comments, we revised our manuscript carefully and thoroughly, particularly on language issues from co-authors' efforts and few logic issues. Since our manuscript has been sent for a language proof before submission, we did not send our manuscript again, but combine co-authors' efforts on language. Please see, below, our point-to-point response.

**Suggestion and comments from referees** are marked in **Black**.

**Responses to referee's comments** are labelled in **blue**.

**Cited changes made in the manuscript** are marked in **red**.

Please do not hesitate to let us know if you have further questions and/or comments.

Sincerely,

Xiaolu Tang, Shaohui Fan and Wunian Yang, on behalf of all co-authors.

Comments to the Author:

Dear Authors,

One reviewer reviewed the revised version. The reviewer agreed that you made efforts to improve the manuscript but had some minor suggestions that you should follow during the final version. The reviewer also thought the paper has some linguistic problems. Please ask a native speaker to check the manuscript and correct these problems before the paper can be accepted.

Response: thank you very much for your efforts again. Since our manuscript has been sent for a language proof before submission, we did not send our manuscript again, but combine co-authors' efforts on language. Please see, below, our point-to-point response.

Suggestions for revision or reasons for rejection (will be published if the paper is accepted for final publication)

Tang et al presented to the carbon science community an interesting dataset of RH, which may be helpful for benchmarking terrestrial biogeochemical models. They also conducted a number of statistical comparisons with other datasets, including the Hashimoto RH, and the TRENDY model outputs, and found some very interesting and logically sound patterns. Overall, I agree with the previous reviewers that this study merits publication. And the authors have worked hard to address the concerns raised in the first 3 reviewers. That being said, however, I still think the paper has some linguistic problems (more than grammar and syntax), which need to be carefully attended before the paper can be accepted. I will give my details below.

Response: thank you very much for your comments and suggestion to improve the manuscript. As suggested, we revised our manuscript carefully again with a focus on language problems. Please see the point-to-point change and the revised manuscript with changes tracked.

P1, L21-22: "high variations and uncertainties of RH existing in global carbon cycling models require an urgent development of data-derived RH dataset". Although I understand the value of a data-derived RH dataset, I don't think the logic here is well established. For instance, I can equally say a more mechanistically based modeling approach is more urgently needed. What a data-derived RH dataset will do is to provide a different angle to quantify the RH besides the biogeochemical modeling approach.

Response: thank you for the good suggestion. We agree with the reviewer and our RH dataset could provide another different angle to quantify RH. We revised the manuscript as:

"However, high variations and uncertainties of RH existing in global carbon cycling models require RH estimates from different angles, e.g. a data-driven angle."

P2, L51-2: "Although it is widely recognized that warming enhances CO2 release from soils, the magnitude of release is uncertain due to variations in the temperature sensitivity of soil organic matter decomposition". This statement is generally right, but insufficient. Because when talking about climate responses (aka warming here), the uncertainty involves both temperature and moisture, with all other granularities under the hood. I believer, this sentence need be revised to better connect with the next sentence.

Response: thank you for the suggestion. As suggested, we revised the manuscript as:

"In addition, environmental drivers of RH, e.g. temperature and soil moisture, are still undergoing changes under climate warming and can affect RH individually or interactively."

P3, L66-67: "potential new numerical/algorithmic methods to better quantification and understanding of large-scale soil carbon flux". There are misuses of adverb, and verb here. Please correct.

Response: we are sorry. the verbs "quantification" and "understanding" were correct to "quantify" and "understand", respectively.

P3, L79: "empirically" should be "empirical".

Response: done.

P3, L89 and throughout the paper: "global vegetation models" is not the right jargon, "terrestrial biogeochemical models" or "land biogeochemical models" are more often used.

Response: done as suggested. We replace "global vegetation models" with "terrestrial biogeochemical models".

However, we kept Dynamic Global Vegetation Model from TRENDY model ensembles, because this is the office name and widely used in publications (Ballantyne et al., 2017; Piao et al., 2019; Sitch et al., 2008).

P6, L169: In what way "the other two proxies were controlled"?

Response: We meant "the other two proxies were controlled to remove their confound effects on RH", and we revised as:

"When analysing the partial correlations between RH and the proxy, the other two proxies were controlled to remove their confounding effects on RH."

P7, L218: "reflecting less stress from environmental limitations". This statement is unclear. The first half of the sentence made a contrast, manifesting a gradient-like transition, but this statement does not describe this clearly.

Response: sorry for the unclear statement. We revised the sentence as:

"reflecting a higher resource limitation in high latitude areas and a less resource limitation in low latitude areas."

P8, L286: "than RH from satellite-driven estimates", seems missing a word like "more" or "higher".

Response: sorry, a "higher" was added.

P12, L360: "negatively feeding back to future climate change". I was not able to figure out the logic behind this inference.

Response: we remove the sentence.

P12, L374-375: The sentence should be better broken into two, because it is mumbling currently.

Response: we reduce the long sentence into two as below:

"Compared to linear regression analysis for predicting soil respiration (as no such global RH predictions were previously available for comparison) with model efficiencies of <50% (Bond-Lamberty and Thomson, 2010; Hashimoto et al., 2015; Hursh et al., 2017), RF algorithms achieved a higher model efficiency of 50% in this study. In addition to a feature selection according the importance value of each variable and avoiding overfitting (Bodesheim et al., 2018; Jian et al., 2018), RF could improve RH modelling accuracies and reduce uncertainties."

P12, L376-377: "This developed RH dataset provides several advantages to the estimation of global RH". The advantage should be established within a comparison, however, what are compared here is unclear. After all, you don't want to people to guess, and guessing is prone to misunderstanding.

Response: thank you. We meant advantages compared to other studies and revised the manuscript as:

"This developed RH dataset provides several advantages to the estimation of global RH compared to previous studies, e.g. Hashimoto et al. (2015) and Konings et al. (2019)."

Section 4.5: Overall the authors did discuss the limitations of this study. But I think if they formulate this alternatively by stating what use can be made with this dataset will be more helpful. Of course, it is up to the authors' decision.

Response: after discussing with co-authors before submission and the suggestion from one reviewer during the review process, the co-authors and reviewer suggested a section on the limitation of this study.

[revised manuscript text omitted]